# New Targeted Agents in Acute Myeloid Leukemia: New Hope on the Rise

**DOI:** 10.3390/ijms20081983

**Published:** 2019-04-23

**Authors:** Stephan R. Bohl, Lars Bullinger, Frank G. Rücker

**Affiliations:** 1Department of Internal Medicine III, University Hospital Ulm, 89081 Ulm, Germany; stephanrm.bohl@gmail.com; 2Department of Hematology, Oncology and Tumorimmunology, Charité University Medicine, 13353 Berlin, Germany; lars.bullinger@charite.de

**Keywords:** AML, targeted therapy, FLT3, IDH1/2, BCL2, hedgehog, immunogenic treatment

## Abstract

The therapeutic approach for acute myeloid leukemia (AML) remains challenging, since over the last four decades a stagnation in standard cytotoxic treatment has been observed. But within recent years, remarkable advances in the understanding of the molecular heterogeneity and complexity of this disease have led to the identification of novel therapeutic targets. In the last two years, seven new targeted agents (midostaurin, gilteritinib, enasidenib, ivosidenib, glasdegib, venetoclax and gemtuzumab ozogamicin) have received US Food and Drug Administration (FDA) approval for the treatment of AML. These drugs did not just prove to have a clinical benefit as single agents but have especially improved AML patient outcomes if they are combined with conventional therapy. In this review, we will focus on currently approved and promising upcoming agents and we will discuss controversial aspects and limitations of targeted treatment strategies.

## 1. Introduction

Acute myeloid leukemia (AML), a genetically heterogeneous disease characterized by the accumulation of acquired genetic changes in hematopoietic progenitor cells, still remains a therapeutic challenge [1]. The median age at diagnosis is about 70 years with a 5-year survival rate of 40% for younger patients (18–60 years) and just 10% for patients above the age of 60 years [1]. Despite improvements in understanding of the molecular biology of the disease, over the last four decades treatment has changed minimally and outcome remains poor for the majority of patients [2]. Until recently, most patients have been treated if eligible with comparable cytarabine/anthracycline-based chemotherapy regimens or with hypomethylating agents (HMA) if declared unfit for intensive treatment. However, following this long stagnation in the anti-leukemic drug development process, the clinical options are now changing fast with new hope on the horizon. This is supported by the progress of unravelling the pathogenesis based on improved sequencing techniques leading to a better understanding of genetic driver mutations in AML and the identification of new molecular markers [3]. Today AML is risk stratified into favourable, intermediate- and high-risk groups based on (cyto-)genetic alterations. In addition, the identifications of recurrent mutations like *NPM1*, *FLT3*, *CEBPA* and *IDH1/IDH2* have updated prognosis and guide AML therapy [4]. Over the last couple of years, several mutation-targeted agents acting on the oncogenic effector *FLT3* (found in roughly 25% of AML patients [3] have been developed and led to promising results in clinical trials. In 2017, the US Food and Drug Administration (FDA) as well as the European Medicines Agency (EMA) approved the first tyrosine kinase inhibitor (TKI) midostaurin in combination with chemotherapy for *FLT3*-mutated AML based on data of a large randomized phase 3 study [5]. Besides, all transretinoic acid (ATRA) for the treatment of retinoic-acid receptor rearranged acute promyelocytic leukemia, midostaurin was the first drug approved in a genetic-specific, non-acute promyelocytic leukemia manner. Recently, additional mutation-specific targeted agents followed midostaurin in the daily clinical use including another FLT3 inhibitor gilteritinib [6], as well as compounds targeting mutant isocitrate dehydrogenase (IDH), enasidenib and ivosidenib [7,8]. In addition to these mutation-specific approvals, two other targeted novel agents have been FDA approved, venetoclax and glasdegib, disrupting anti-apoptotic or cell maintenance pathways without damaging DNA, respectively [9,10,11]. Finally, the previously approved but later withdrawn anti-body drug conjugate (ABDC) gemtuzumab ozogamicin (GO) received FDA re-approval for CD33 positive AML [12]. Therefore, it seems that the time of “one hits them all” frontline intensive chemotherapy (IC) for AML is finally changing, since these new drugs are starting to reshape the therapeutic strategies in AML towards precision medicine approaches (Figure 1).

## 2. Anti-Body Drug Conjugate (ABDC)

ABDCs combine cytotoxic agents with a targeted approach, which by attaching the cytotoxic drug to an antibody can lead to an increased dose intensity with reduced toxicity. Over the last decade, primary focus has been set on the sialic acid-binding immunoglobulin-like lectin CD33 as a potential target in AML. However, surface CD33 is highly variable among myeloid blasts [13]. In addition to leukemic blasts CD33 expression is generally restricted to myeloid progenitors and more differentiated blood cells. The intracellular phosphorylation of the cytoplasmic tail of CD33 creates docking sites for recruitment and activation of tyrosine phosphatases or suppressor of cytokine signalling. First clinical trials of humanized antibodies targeted against CD33 showed only limited activity in AML [14]. Because CD33 is internalized rapidly when bound by antibodies, conjugating cytotoxic molecules to the antibodies improved the anti-leukemic efficiency.

### Gemtuzumab Ozogamicin (GO)

For GO the CD33 antibody is conjugated to the DNA intercalating antibiotic calicheamicin via a hydrolysable linker [13]. Once the drug conjugate is internalized into the cell, calicheamicin is released generating single and double strand breaks with subsequent cellular death. GO received accelerated FDA approval in 2000 as a novel AML monotherapy based on three single arm phase 2 trials. But for full regulatory approval, it was required to confirm the clinical benefit in further phase 3 trials. In the first published phase 3 clinical trial (SWOG S0106) including 595 younger (<60 years) de novo AML patients were randomized to receive IC (daunorubicin and cytarabine) plus/minus GO 6 mg/m^2^ on day 4 [15]. This study failed to show a clinical benefit for GO and was stopped early because of an interim analysis having demonstrated that adding GO to IC did not lead to improved complete remission (CR) (70% vs. 69%) and/or overall survival (OS) rates (5-year survival rate 46% vs. 50%) but was even associated with a higher mortality rate in induction treatment (5.5% vs. 1.4%; of note, the early death (ED) rate in the control arm of this cooperative group trial was remarkably low) [15]. Based on these negative results including emerging safety concerns like for example, hepatic sinusoidal obstructive disease, GO was withdrawn from the market in 2010. However, several subsequent trials investigated different schedules of GO to reduce toxicity and to maximize efficacy like for example, the ALFA-0701 trial [13]. Within this phase 3 trial, 278 de novo AML patients (50–70 years) were randomized to IC (daunorubicin and cytarabine) plus/minus a fractionated GO induction regime (3 mg/m^2^ on day 1, 4 and 7 during induction plus additional dosing during consolidation) [12]. Although the CR rates were similar between the two arms, IC plus GO provided a significantly improved median event free survival (EFS) (19.6 vs. 11.9 months, *p* = 0.00018) and median OS (34 vs. 19.2 months, *p* = 0.046). The safety profile analysis showed a prolonged recovery for neutrophils and platelets but no increase of hepatic sinusoidal obstructive disease. Subgroup analysis showed that clinical benefit was restricted to cytogenetic favourable and intermediate risk groups [12]. A meta-analysis of five phase 3 trials comprising 3325 AML patients disclosed a significant reduction of relapse rates and an improved OS without increased toxicity for GO treatment [16]. Again, the benefit was restricted to cytogenetic favourable and intermediate risk groups but also to patients receiving a lower dose of GO (3 mg/m^2^ instead of 6 mg/m^2^). Based on these results GO received full FDA and EMA approval for frontline and relapsed therapy of CD33 positive AML in 2017 and 2018, respectively. In a further phase 3 trial for *NPM1* mutated de novo AML (*n* = 588) randomized to IC (idarubicin, cytarabine, etoposide and ATRA) plus/minus GO 3 mg/m^2^ on day 1 there was no difference in cCR rate after induction therapy (88.5% versus 85.3%, *p* = 0.28) but the GO treatment was associated with a higher ED rate (7.5% vs. 3.4%; *p* = 0.02), particularly in patients aged over 70 years. In patients who achieved a composite CR (cCR, defined as CR plus complete response with incomplete hematologic recovery (CRi)) after induction therapy, those treated in the GO arm exhibited a significantly lower cumulative incidence of relapse (*p* = 0.018) [17]. These results demonstrate that GO administered in a fractionated dosing schedule has an improved safety profile without compromising clinical efficacy. However, the risk of hepatic sinusoidal obstructive disease has to be kept in mind and additional hepatotoxic medications should be avoided. Next to low CD33 expression as seen in adverse cytogenetic risk group, the multidrug resistant P glycoprotein, a transmembrane glycoprotein that pumps several anti-leukemic agents out from cells, seems to affect GO efficacy and may cause resistance [18].

## 3. FLT3-Inhibitors

FLT3 (fms related tyrosine kinase 3), a cytokine receptor (CD135) belonging to the receptor tyrosine kinase class III, is expressed mainly on hematopoietic cells [19]. FLT3 takes a pivotal role in myeloid and lymphoid cell proliferation and survival [20]. In AML, two mutations of the *FLT3* gene are recurrently found: (i) *FLT3* internal tandem duplications (*FLT3*-ITDs) of the juxtamembrane domain occurring in around 25% of patients) [21] and (ii) point mutations in the tyrosine kinase activating loop of the kinase domain *FLT3*-TKD (typically at codon D835) in about 5–10% of patients [21]. Both genetic aberrations lead to constitutive activation of the kinase promoting cell growth, survival and antiapoptotic signalling (Figure 1). *FLT3*-ITDs are associated with adverse prognosis due to a high relapse rate, in particular in case of a high mutant to wild-type allele ratio and/or insertion site in the beta1-sheet of the tyrosine kinase domain-1 [22], while the impact of *FLT3*-TKD mutations remains less clear in AML patients [23].

Today, several kinase inhibitors are being explored in clinical trials since 2002. First generation FLT3 inhibitors like midostaurin and sorafenib were not designed to specifically target FLT3 and also show activity against KIT, PDGFR and VEGFR, thereby leading to more off-target associated toxicity and side effects [24]. Thus, it was a major challenge to identify a clinical active and tolerable dose providing sufficient kinase inhibition throughout the dosing interval [25].

### 3.1. Sorafenib

Sorafenib, a pan kinase inhibitor, has been approved in several solid malignancies. In an early phase clinical trial, sorafenib combined with idarubicin and high dose cytarabine in younger de novo AML patients provided a CR rate of 93% and a 1-year survival rate of 74% in *FLT3*-ITD positive AML patients [26]. Sorafenib was generally well tolerated with diarrhoea and rash being the most frequent adverse events. In the randomized phase 1 SORAML trial, 276 newly diagnosed AML patients (<60 years) were allocated to receive IC plus either sorafenib or placebo. The authors reported a significantly improved 3-year EFS (40% sorafenib vs. 22% placebo arm, *p* = 0.013) independent of *FLT3* mutation. This may be caused by off-target effects of sorafenib. Nevertheless, the prolonged EFS did not lead to a benefit in OS [27] because after relapse, patients of the placebo cohort exhibited a longer OS compared to the sorafenib cohort (26 months vs. 7 months, *p* = 0.039). The authors suggested that salvage treatment, primarily allogeneic stem cell transplantation (HSCT), may not have been equally potent in patients relapsing after placebo or sorafenib therapy, since sorafenib may select for resistant AML subclones In a lower intensity treatment approach azacitidine plus sorafenib demonstrated valid clinical activity in r/r *FLT3*-ITD positive AML [overall response rate (ORR) 46%, 16% CR] and elderly de novo *FLT3*-ITD positive AML patients (ORR 78%, 26% CR) [28,29]. Based on encouraging results of single arm studies investigating sorafenib maintenance treatment after HSCT [30,31], sorafenib maintenance post HSCT in *FLT3*-ITD positive AML was explored in a double-blind placebo control trial. After a median follow up of 41.8 months the median EFS was 30.9 months in the placebo group whereas it was not reached in the sorafenib group translating into a 2-year relapse free survival of 53% for the placebo versus 85% for the sorafenib group (*p* = 0.0135) [32].

### 3.2. Midostaurin

Midostaurin is another first-generation multi-kinase inhibitor [33]. Weinberg and colleagues demonstrated a FLT3 inhibitory activity of midostaurin by performing a drug screen [34]. Based on monotherapy phase 1 trials further studies were initiated combining midostaurin with IC. In 40 younger AML patients (<60 years) midostaurin plus IC provided an overall CR rate of 80% [90% in *FLT3*-ITD, 74% in *FLT3*-wild type (WT)] but had no impact on OS [35]. In the following placebo-controlled phase 3 trial 717 patients with newly diagnosed *FLT3* mutated AML (*FLT3*-ITD and *FLT3*-TKD) were randomized to IC plus/minus midostaurin (RATIFY) [5]. Additional midostaurin (50 mg orally twice daily) did not lead to a higher CR rate but significantly improved OS (4-year survival probability, 51% vs. 44%; *p* = 0.009) and EFS (4-year survival probability, 28% vs. 21%; *p* = 0.002) [5]. Generally, midostaurin was well tolerated with febrile neutropenia and gastro-intestinal adverse events being the most common side effects and in just 3.1% of patients adverse events led to an interruption [5]. The OS benefit for midostaurin remained even after censoring for HSCT linked to a deeper response rate, like minimal measurable residual disease (MRD) negativity, as just recently proved by Lewis and colleagues using a next generation sequencing (NGS) based MRD analyses of patients treated within the RATIFY trial [36,37].

The results from the RATIFY/Alliance 10603 trial finally led to the approval of midostaurin for the treatment of *FLT3* mutated AML in 2017. However, the impact of maintenance treatment with midostaurin on overall outcome remains unclarified [38]. A phase 1/2 trial explored the combination of midostaurin with low intensity therapy like azacitidine irrespective of *FLT3* mutation status in untreated and r/r AML elderly patients [39]. In untreated *FLT3*-ITD positive patients a response rate of 33% was achieved with a significantly longer response duration of 31 weeks compared to *FLT3*-ITD positive patients previously exposed to other FLT3 inhibitors (31 vs. 16 weeks, *p* = 0.05). Based on its broad kinase inhibitory function, currently a placebo controlled randomized phase 3 trial also investigates the impact of adding midostaurin to IC in *FLT3*-WT AML patients (NCT03512197) (Table 1).

Unlike the first generation of FLT3 inhibitors, newer agents, such as quizartinib, crenolanib and gilteritinib, are more selective and more potent inhibitors of FLT3 [2].

### 3.3. Quizartinib

In contrast to midostaurin, quizartinib is a highly selective second generation receptor tyrosine kinase inhibitor (RTK) designed to target FLT3 and few other kinases like KIT [40], thereby showing only limited toxicity and less off-target effects. Quizartinib’s side effects are usually mild but QT interval prolongation occurs commonly at higher doses (90–135 mg/day). However, in trials with lower dose rates (30–60 mg/day) there were comparable response rates but less QT interval prolongation [41,42]. This might be associated with the prolonged half-life of >24 h of quizartinib leading to a continuous FLT3 inhibition [41]. However, quizartinib does not show activity against *FLT3*^D835^ mutated AML [43].

In a large phase 2 trial assigning 333 r/r AML patients, quizartinib as a single agent achieved a cCR rate of 50% in *FLT3*-ITD positive AML but a CR rate of only 3% due to ongoing treatment-related cytopenia likely caused by additional inhibition of KIT [44]. Quizartinib also exhibited activity in *FLT3* WT AML achieving a cCR of >30%. Albeit the median time of response duration was less than 3 months, 35% of younger patients were able to undergo HSCT [44]. Interestingly, a considerable fraction (47%) of responding patients had a hypercellular bone marrow with a persistent *FLT3*-ITD/*FLT3*-WT allelic burden despite of blast reduction to <5% [45]. It was considered that hypercellular bone marrow was due to the induction of terminal granulocytic differentiation [46]. Preliminary results of a randomized phase 3 study (QuANTUM-R) investigating 367 r/r *FLT*-ITD positive AML patients receiving either quizartinib or salvage chemotherapy (IC and low intensity) showed a significantly improved median OS for quizartinib (6.2 vs. 4.7 months; *p* = 0.017) and an improved cCR rate (48% vs. 27%, *p* = 0.0001) [47,48]. An interim analysis of a phase 1/2 trial of quizartinib combined with azacitidine or low dose cytarabine in 52 AML patients (irrespective of *FLT3* mutations) demonstrated an ORR of 67% [48]. The 1-year survival rate was significantly higher for the azacitidine arm compared to the low dose cytarabine arm (72% vs. 32%, *p* = 0.027). A placebo controlled double blind trial (QuANTUM-FIRST) investigating quizartinib combined with IC for newly diagnosed *FLT3*-ITD positive AML is ongoing (NCT02668653) (Table 1).

### 3.4. Crenolanib

Crenolanib is a second generation RTK inhibiting *FLT3*-ITD and -TKD mutations. Crenolanib was well tolerated at a dosage of 200 mg/m^2^/day three times a day with primarily gastrointestinal adverse events like abdominal pain or nausea and showed only minimal potential for QT interval prolongation in r/r AML patients [49]. In a phase 1 trial of *FLT3*-ITD positive AML crenolanib as single agent achieved an ORR of 50% in 18 patients naive for FLT3 inhibitor treatment and of 31% in 36 patients previously treated with a FLT3 inhibitor like sorafenib or quizartinib [50]. Currently, several clinical trials are exploring crenolanib in relapsed as well as in frontline setting combined either with IC or hypomethylating agents (Table 1). Preliminary results of a phase 2 trial in 28 r/r *FLT3* mutated (ITD and TKD) AML (16 with prior FLT3 inhibitor exposure, 20 combined with IC, 8 combined with azacitidine) showed an ORR of 46% (including 10 cCR). The median OS for patients having received less than two prior therapies was 6.2 months versus 1.5 months (*p* = 0.0002) for patients having received more than two prior therapies [51]. Wang and colleagues presented results of younger *FLT3* mutated (ITD and TKD) AML patients (<60 years) treated with crenolanib in combination with IC: The overall cCR rate was 83% with a *FLT3* mutation clearance in 91% of 23 evaluable patients. Among patients who achieved a cCR, 94% became MRD negative after just one treatment cycle. The OS after a median follow-up time of around 18 months was 79% [52]. Multiple phase 3 trials are currently ongoing, including comparisons of crenolanib versus midostaurin (NCT02298166, NCT03258931) (Table 1).

### 3.5. Gilteritinib

Gilteritinib is another potent and selective dual FLT3 (to a lesser extent to *FLT3*-TKD than -ITD) and AXL inhibitor. AXL is another RTK that promotes proliferation and survival of AML cells [53]. Taken only once a day at a dosage of 120 mg, gilteritinib demonstrated an encouraging cCR of 30% as single agent in 252 r/r AML patients (with a cCR rate of 41% and a CR rate of 11% in 169 *FLT3* ITD and TKD mutant patients) [54]. The response rates for patients with only *FLT3*-ITD and those with *FLT3*-ITD and *FLT3*^D835^ mutations were identical. The most common non haematological adverse events were diarrhoea and hepatic enzyme elevation. The phase 3 ADMIRAL trial assessing oral gilteritinib 120 mg per day versus salvage chemotherapy in adult r/r *FLT3* mutated AML patients led to an FDA approval for gilteritinib. This was based on an interim analysis demonstrating a cCR of 21% with a duration of 4.6 months. Patients who achieved a cCR had a median time to response of 3.6 months [6]. Furthermore, gilteritinib is currently studied as upfront treatment versus midostaurin in combination with IC and as maintenance therapy following induction/consolidation treatment in first remission (NCT02236013, NCT 0292762) (Table 1).

In AML, *FLT3* mutations are associated with a poor prognosis, however, the development of these new targeted agents has improved outcome of this AML subtype. Midostaurin being the first TKI approved for AML therapy in first line raised the bar for newer and more potent agents seeking into frontline therapy since the control arm has to include midostaurin. In addition, it will be interesting to see if more specific second generation FLT3 inhibitors like quizartinib, crenolanib and gilteritinib will lead to an improved outcome compared to midostaurin. Despite the development of more potent FLT3 inhibitors, resistance and subsequent relapse of AML is still a major challenge being currently under investigation [55]. Potential resistance mechanisms include suboptimal drug levels within the bone marrow [56], aberrant signalling bypassing the FLT3 receptor [57], acquisition of *TKD* mutations [58] and activation of alternate signalling pathways [59].

## 4. IDH inhibitors

The isocitrate dehydrogenases IDH1 and IDH2 are ubiquitously expressed enzymes catalysing the oxidative decarboxylation of isocitrate to a-ketoglutarate (aKG) in the cytoplasm and the mitochondria. *IDH1* and *IDH2* are recurrently mutated in about 20% of AML [60]. These mutations encode for neomorphic enzymes hampering the enzymatic activity and conferring the ability to catalyse the conversion of aKG to the oncometabolite R-2-hydroxyglutarate which perturbs DNA and histone methylation in hematopoietic stem cells [61,62] (Figure 1). In AML, *IDH1* and *IDH2* mutations are mutually exclusive and distinct from other mutations like *TET2* or *WT1* [3]. Inhibitors of the mutant IDH enzymes are capable to decrease the total serum level of R-2-hydroxyglutarate, therefore, reducing aberrant histone hypermethylation and inducing myeloid differentiation [63].

### 4.1. Enasidenib

Enasidenib is a bivalent inhibitor of R140Q and R172K mutated IDH2 and has been the first IDH mutation specific inhibitor [64]. Enasidenib generates terminal differentiation of myeloid blasts into neutrophils in vivo. In a phase 1/2 trial, 100 mg enasidenib daily administered orally achieved an ORR of 39% with a cCR of 30% in r/r *IDH2* mutant AML patients [7]. The median OS was 8.8 months and patients who achieved a CR exhibited a median OS of 19.7 months. Generally, enasidenib was well tolerated, treatment-related adverse events were hyperbilirubinemia and thrombocytopenia. Differentiation syndrome occurred in 6% of patients characterized by fever, dyspnoea due to lung infiltrates, pleural effusion and leukocytosis [7]. In a molecular analysis, clinical response to enasidenib was even seen without reduction of the *IDH* mutant allele burden suggesting that the primary mechanism of response appears to be induction of terminal differentiation [65]. The attainment of CR was associated with *IDH2* allele burden reduction and molecular clearance [7]. Based on the observation that responses are seen even in patients with very small *IDH2* mutant clones it has been proposed that enasidenib may have additional paracrine effects on *IDH* WT myeloid blasts [66]. On the other hand, specific co-mutations, especially in *NRAS* or *KRAS* but also *FLT3*, were associated with lower response rates [65]. These results led to the FDA approval of enasidenib in r/r *IDH2* mutated AML patients. In frontline treatment, enasidenib added to IC achieved a cCR rate of 72% as shown in a phase 1 trial of 93 older high-risk *IDH2* mutated AML patients. Of these, 45% of patients who achieved a CR became also MRD negative and 25% showed an *IDH2* mutation clearance [67]. These results demonstrate that enasidenib in combination with intensive chemotherapy achieves robust remission rates, MRD-negative CRs and mutation clearance in an elderly AML population [67]. Currently, the clinical benefit of adding enasidenib to induction, consolidation and maintenance therapy for patients with newly diagnosed *IDH2* mutant AML is under further evaluation in randomized phase 3 trials (NCT03839771, NCT02577406) (Table 1).

### 4.2. Ivosidenib

Ivosidenib is a potent and selective IDH1 mutation inhibitor and has also shown promising results in phase 1/2 trials. The clinical efficacy of ivosidenib was explored in a phase 1 dose escalation study including 258 patients with *IDH1* mutated hematologic malignancies. In 125 r/r AML patients an ORR of 41% was reported including a cCR rate of 30% and a CR rate of 22% [8]. The median OS was 8.8 months for all AML patients and 18 months for patients who achieved a cCR. Frequent adverse events of ivosidenib were leukocytosis, QTc prolongation and differentiation syndrome [8]. The results of this trial led to the FDA approval of ivosidenib in r/r *IDH1* mutated AML patients. In 34 untreated AML patients ivosidenib showed an ORR of 58% including a cCR rate of 42% translating into a median OS of 12.6 months (median follow up: 23.1 months) [68]. Among the patients who achieved a cCR, an *IDH1* mutation clearance in bone marrow was observed in 64%. A phase 1 trial of ivosidenib (500 mg daily) as frontline treatment in combination with IC assigning 60 patients showed a cCR rate of 80%. Notably, 88% of patients who achieved a CR became also MRD negative and 41% showed an *IDH1* mutation clearance [67]. Therefore, ivosidenib seems to achieve reliable response and remission rates as well as MRD negativity in older higher risk *IDH1* mutated AML patients when combined with IC. The clinical benefit of ivosidenib in combinational therapy is currently under further evaluation in randomized phase 3 trials (NCT03173248, NCT03839771) (Table 1).

Both IDH inhibitors exhibit robust biological activity in r/r *IDH1/2* mutated AML and may even enhance MRD negativity in combination with IC as first line treatment. Due to the potential life threatening differentiation syndrome physicians should remain alert, especially in the clinical outpatient setting. Potential resistance mechanisms have already been proposed like clonal evolution or selection of terminal or ancestral clones [69], emergence of second-site *IDH2* mutations [70] and identifications of mutant *IDH* isoform switching either from mutant *IDH1* to mutant *IDH2* or vice versa [71].

## 5. Hedgehog Inhibition

The hedgehog (HH)/Glioma associated Oncogene Homolog (GLI) signalling pathway is essential for embryonic development and usually silenced in adult tissues [72]. HH/GLI aberrant signalling seems also to be pivotal for several cancer hallmarks like genomic instability, proliferative signalling, replicative immortality, tumour invasion-metastasis, inflammation and immune-surveillance evasion mechanisms [73]. Especially cellular self-renewal and evading apoptosis have been part of scientific studies, as they are suggested to cause resistance to chemotherapy and to contribute to cancer stem cell formation [74] (Figure 1). Several studies have indicated that these cancer stem cells are also found in AML [75] and that for leukemia stem cells (LSC) aberrant HH/GLI signalling is critical for survival and expansion [76]. Overexpression of various HH/GLI components have been found in chemotherapy resistant myeloid blasts and subsequent inhibition of the HH/GLI pathway revised the sensitivity to chemotherapy [77]. These pre-clinical results provide the rationale for combining chemotherapy with HH/GLI pathway inhibitor in myeloid malignancies, albeit several clinical trials have failed to demonstrate a clinical benefit of HH/GLI inhibitors in various solid cancers [78].

###  Glasdegib

Glasdegib is a potent and selective oral inhibitor of HH/GLI signalling targeting the essential pathway effector Smoothened (SMO). In several in-vitro studies glasdegib achieved a complete tumour growth arrest as single agent and in combination with chemotherapy and glasdegib also attenuated LSC populations in AML models [79,80]. These results led to an open-label dose finding phase 1 trial in 42 patients with myeloid malignancies (AML, *n* = 28) who were refractory, resistant or intolerant to previous treatments. The maximally tolerated dose of glasdegib was 400 mg once daily with non-hematologic adverse events including dysgeusia, decreased appetite, fatigue and alopecia. Among the 28 AML patients, in 16 (57%) a biological activity of glasdegib was observed with one CR, 4 partial remissions, 4 minor responses and 7 stable diseases [81]. Based on these results a phase 1/2 trial with a maximum dosage of 200 mg was recommended. In the following phase 1/2 trial including previously untreated AML and high-risk MDS patients (*n* = 42) glasdegib was combined with either low dose cytarabine, decitabine or IC. The cCR rates among the different treatment arms were 8%, 28% and 54%, respectively. These treatment approaches led to median OS of 4.4, 11.7 and 34 months (10 patients in the IC arm underwent HSCT), respectively [82]. A subsequent biomarker analysis delineated improved response in patients with *FLT3* mutations compared to *FLT3*-WT (median OS *FLT3* mutated unreached versus 13.1 months; *p* = 0.0036) [83]. In a randomized phase 2 multicentre study of 132 AML or high-risk MDS patients evaluating low dose cytarabine plus/minus glasdegib (100 mg oral) the CR rates were 17% for the glasdegib arm versus 2% for the standard arm translating into a median OS of 8.8 versus 4.9 months (*p* = 0.0004). [11]. Based on this study the FDA approved glasdegib in combination with low dose cytarabine for AML patients unfit for IC. Currently studies are exploring the benefit of glasdegib in combination with IC or AZA versus IC or AZA alone (NCT03416179) (Table 1).

Glasdegib shows a clinical benefit when combined with low dose cytarabine. Since low dose cytarabine is inferior to HMA therapy, it might be questioned whether glasdegib will meet the expectations in clinical routine if current studies exploring glasdegib in combination with IC or AZA would fail to show any clinical benefit. Moreover, little is known about potential resistance mechanism to glasdegib but it seems that the GLI3 transcriptional repressor determines response to SMO inhibition [84].

## 6. BCL2 Inhibition

In MDS and AML cells conventional cytotoxic chemotherapy and HMA treatment induce mitochondrial-mediated apoptosis [85,86], a form of programmed cell death in response to cellular stress regulated by the BCL2 protein family (Figure 1). BCL2 family proteins are characterized by the presence of at least 1 of 4 BCL2 homology domains (BH1–4) and are classified into pro- and anti-apoptotic proteins. The pro-apoptotic proteins are capable of either directly activating effector proteins or antagonizing antiapoptotic proteins of the BCL2 family [87], thereby leading to an activation of caspase proteases [88]. BCL2 protects cells from diverse stress including chemotherapy. Overexpression of anti-apoptotic BCL2 proteins such as BCL2, BCL2L1 and MCL1 is widely associated with tumour initiation, progression and chemo resistance in AML [89]. Therefore, BCL2 represents a promising therapeutic target.

### Venetoclax

Venetoclax, a highly selective oral BCL2 inhibitor lacking affinity for BCL-XL or MCL-1, has been shown to induce apoptosis in AML cell lines and primary patient samples in-vitro and in mouse xenograft models [90,91]. Single agent venetoclax has been investigated in clinical trials in r/r AML and achieved an ORR of 19% (15% cCR) [92]. Due to potential tumour lysis syndrome caused by large amounts of decayed tumour cells and potentially leading to multi-organ failure and even death, as seen in chronic lymphocytic leukemia, a daily dosing ramp up of venetoclax was executed until 800 mg per day. Common drug related adverse events were nausea, diarrhoea and febrile neutropenia. The responses were only short lasting with a median progression free interval of just 2.5 months. In a phase 1/2b trial of unfit AML patients venetoclax 600 mg in combination with low dose cytarabine achieved a cCR rate of 54% [9]. In contrast to single agent venetoclax, response duration in combination with low dose cytarabine was enhanced with a median time of 8.1 months resulting in a median OS of 10.1 months [9]. Intermediate risk cytogenetics exhibited a higher response rate compared to adverse risk cytogenetics (cCR 63% vs. 42%) coming along with an increased survival (OS 15.7 vs. 4.8 months) [9]. Venetoclax was also combined with HMAs in a dose escalation study for older AML patients (>65 years) not eligible for IC. Venetoclax was administered at 400, 800 or 1200 mg daily in combination with either azacitidine or decitabine [10]. The cCR rate was 67% irrespective of venetoclax dosage with a cCR rate of 73% for the venetoclax 400mg plus HMA arm. Patients with adverse-risk cytogenetics and those above the age of 75 years exhibited compared to historic controls higher cCR rates of 60% and 65%, suggesting venetoclax/HMA combination might overcome the poor prognosis of these subgroups. Similar to the low dose cytarabine trial, the median time of response duration for patients in cCR was 11.3 months with a median OS of 17.5 months; for the 400mg venetoclax cohort median OS was not reached at the time point of analysis [10]. The combination of venetoclax and HMA suppresses oxidative phosphorylation by disruption of the tricarboxylic acid cycle as manifested by decreased aKG and increased succinate levels, thereby eradicating LSCs [93]. In both trials no tumour lysis syndrome was observed and most responses occurred within two cycles. Due to these marked results venetoclax received FDA approval for combination with low dose cytarabine and HMAs. Preliminary data of venetoclax added to IC in heavily pre-treated r/r AML patients showed a cCR rate of 73% with a 6-month survival rate of 67%, whereas the median response duration was not yet reached [94]. Ongoing randomized phase 3 trials are assessing the potential benefit of venetoclax in the low intensive treatment setting (NCT02993523, NCT03069352) (Table 1).

In contrast to single agent use, in older AML patients venetoclax displays strong biological activity when combined with either low dose cytarabine or HMAs offering a quite competitive therapy option especially for intermediate and low risk AML patients. Based on these results venetoclax shows hope for elderly AML patients with previously limited treatment options and frustrating clinical outcome. The combination of venetoclax and a targeted agent like FLT3-inhibitor may even improve outcome of high risk *FLT3*-ITD positive AML patients. Resistance mechanisms to venetoclax in AML are part of ongoing research; both mutational driven mechanisms and apoptotic evasion mechanism have already been proclaimed [95,96]. For example, in chronic lymphocytic leukemia patients acquiring the novel BCL2^G101V^ mutation exhibit resistance to venetoclax [97]. However, preclinical data showed that resistance to venetoclax may be overcome by an MCL-1-inhibitor, which is currently being investigated in early clinical trials [98,99].

## 7. Upcoming Agents

In addition to the targeted drugs discussed above, several promising novel agents are under way for AML treatment including inhibitors of epigenetic BET regulators [bromodomain (BRD) and extraterminal (BET) proteins], disrupters of telomeric silencing 1-like (DOT1L) and lysine specific demethylase (LSD1), as well as a KMT2A-menin- inhibitors. Furthermore, immunogenic drugs like CD33/CD3 or CD123/CD3-bispecific T-cell engaging (BiTE) antibody construct and chimeric antigen receptor (CAR)-T-cells might be promising approaches for future AML treatment. The discovery that epigenetic readers of the BRD and BET protein family seem to be crucial for leukemic blasts by transcription of oncogenic c-MYC led to rapid development of BET inhibitors currently explored in clinical trials. Data for AML are limited so far but among 36 r/r AML patients the BET inhibitor OTX015 showed modest response with an ORR of 13% [100]. Further clinical trials involving other BET inhibitors are currently ongoing (NCT01943851, NCT02391480).

LSD1, a histone H3K4me1/2 demethylase, is critical for hematopoietic differentiation and is overexpressed in multiple cancers. Recent studies suggest that inhibition of LSD1 may reactivate the all-trans retinoic acid receptor pathway in AML [101]. Currently, clinical trials are investigating LSD1 inhibitor as monotherapy and in combination with azacitidine (NCT02177812, NCT02929498). DOT1L interacts with lysine methyltransferase 2A (KMT2A) fusion proteins, catalysing aberrant H3K79 methylation of KMT2A target gen loci conferring stem cell like properties and leukemogenesis [102]. AML with KMT2A fusion proteins accounts for about 5–10% in adults and responds poorly to current treatment approaches [103]. Pinometostat, a DOT1L inhibitor, showed antileukemic effect in KMT2A-rearranged AML with an ORR of 12% (6 of 49 patients) including two CRs and resolution of leukemia cutis (*n* = 3) [104]. Pinemostat is currently explored in combination with IC in KMT2A-rearranged AML (NCT 03724084). Another interesting target in KMT2A-rearranged AML is the protein menin that interacts with KMT2A fusion proteins and seems to be an oncogenic cofactor in KMT2A driven leukemogenesis [103]. Based on promising pre-clinical data the menin inhibitor KO-539 received FDA clearance for clinical investigation and a phase 1 clinical trial will be initiated within this year. A major therapeutic pillar of HSCT in AML relies on the graft versus leukemia effect suggesting that earlier immunogenic driven therapy might be a promising approach for AML patients. BiTE are bispecific antibodies directing cytotoxic T-cells against cancer cells. The BiTE antibody blinatumomab targeting CD19 and CD3 is already approved for acute lymphoblastic leukemia (ALL) treatment. For AML it is not surprising that CD33 seems to be a promising target being expressed on most AML cases [105]. A current dose escalating phase 1 study provided provisional data of 35 r/r AML patients: the cCR rate was 12% with a serious adverse events rate of 66% including cytokine release syndrome but no treatment related death [106]. In addition to CD33, another target for BiTE antibody treatment is CD123, the alpha-chain of the interleukin-3 receptor being highly expressed on AML blasts and LSCs. In a phase 1 study for r/r AML and MDS patients flotetuzumab a novel T-cell redirecting CD123/CD3 BiTE displayed an ORR of 43% (28% cCR) with a grade ≥3 drug-related adverse event rate of 44%, mostly cytokine release syndrome [107]. Another therapeutic approach relies on autologous cytotoxic T-cells that are genetically engineered to produce an artificial T-cell receptor targeting cancer cells, CAR-T-cells. This highly anticipated treatment procedure has already been approved for r/r ALL in young adults and r/r diffuse large B-cell lymphoma. In contrast to B-cell malignancies only few clinical trials have investigated CARs for AML. This is partly due to the heterogeneity of myeloid LSCs [108]. In addition to CD33, several targets are currently under clinical and pre-clinical investigation like for example, CD123, CLL1, Lewis Y, NKG2D and FLT3. In a first in human phase 1 study CLL1-CD33 compound CAR-T-cells led to a CR of a 6 year old female patient subsequently followed by HSCT resulting in sustained molecular remission [109]. The fact that thirteen clinical trials are currently investigating CAR-T-cells in AML treatment underlines the highly anticipated expectations behind this treatment approach.

## 8. Future Perspectives

Despite the compelling progress in AML treatment and the development of new targeted therapies, most adult patients will still die from the disease. Today, the most curative treatment approach remains HSCT but is associated with a high therapy related mortality and most AML patients are not eligible for this intensive treatment route. This emphasizes the need for more targeted and less toxic treatment. The identification of molecular aberrations guided the development of mutational driven therapies for *FLT3* and *IDH1/2* mutated AML. For fit AML patients IC remains the treatment back bone and for *FLT3* mutated patients accounting for about 25% the addition of midostaurin is now standard of care. The benefit of additional enzyme inhibition in combination with IC in first line treatment for *IDH1/2* mutated AML needs to be further evaluated. In addition, it has to be taken into account that only a small fraction of AML patients harbour these mutations (8% *IDH1* and 12% *IDH2*). Moreover, new agents targeting both mutant IDH enzymes like AG-881 (NCT02492737) may be more relevant for future clinical use. Further questions to be addressed are combinations, scheduling and timing of targeted maintenance therapy. Still challenging are r/r AML, especially if they occur during or immediately after targeted therapy. For *FLT3* mutated AML, second generation TKIs like gilteritinib, crenolanib and quizartinib have shown benefit for r/r *FLT3* mutated AML facilitating HSCT in CR. While mutational driven targeted therapies are now available for about 45–50% of AML patients (25% *FLT3*, 8% *IDH1*, 12% *IDH2* mutated) half of the AML patients do not yet have the opportunity to be treated with targeted therapies. While the benefit of midostaurin for *FLT3* WT AML patients is currently being investigated, one additional option for these patients might be the addition of leukemia specific pathway inhibition by venetoclax or glasdegib to low intensive therapy, in particular in older AML patients. The impact of these drugs combined with IC on outcome of younger AML patients will be seen over the next years. The combination of mutational driven and pathway driven agents may give us even more therapeutic options and the upcoming new agents provide additional hope. However, the progress being accomplished through these new drugs comes with an economic burden. This has to be critically rendered and kept in an acceptable setting to make novel treatment strategies accessible for as many patients as possible.

## 9. Conclusions

In conclusion, targeted therapy seems to be the future for AML management, in particular for unfit patients and we may hope that someday we do not have rely on IC and allogeneic HSCT alone anymore. The FDA approval of midostaurin, gilteritinib, enasidenib, ivosidenib, glasdegib, venetoclax and gemtuzumab ozogamicin within one year seems to be just the beginning and new pathway inhibiting compounds as well as immunogenic based treatment strategies are already under further clinical investigation. It will be interesting to pursue how the combination of certain targeted agents will further improve the outcome of AML patients.

## Figures and Tables

**Figure 1 ijms-20-01983-f001:**
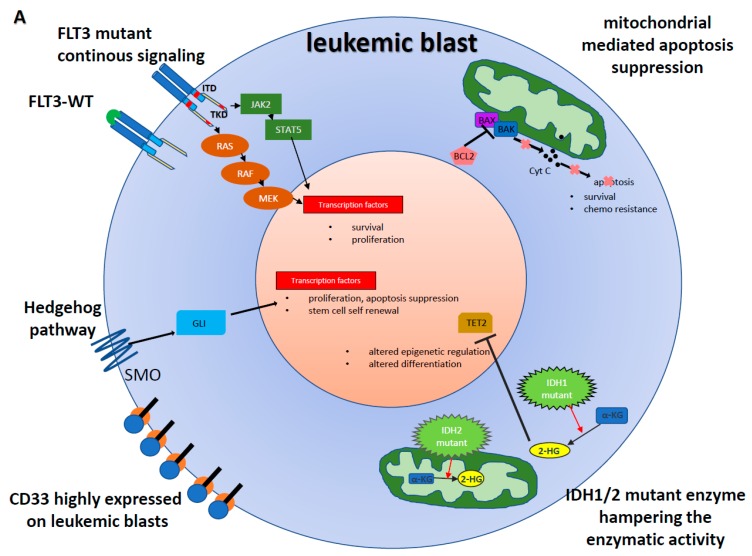
(**A**) Schematic illustration of aberrant and potentially druggable signalling in leukemic blasts leading to cellular proliferation and survival advantage in acute myeloid leukemia (AML). (**B**) Targets of new AML treatment agents inhibiting/blocking impaired cellular pathways and inducing leukemia cell death.

**Table 1 ijms-20-01983-t001:** Selected trials currently enrolling patients for AML featuring targeted agents.

Agents	Investigation	Phase	Identifier
Gemtuzumab Ozogamicin	Liposome-encapsulated daunorubicin-cytarabine and GO in treating patients with r/r AML or high-risk MDS	1	NCT03672539
	Fractionated GO in treating MRD measurable residual disease in participants with AML	2	NCT03737955
Sorafenib	Sorafenib + busulfan and fludarabine conditioning in r/r AML undergoing stem cell transplantation	1/2	NCT03247088
	Sorafenib plus azacitidine in AML/MDS patients with *FLT3*-ITD mutation	2	NCT02196857
Midostaurin	Midostaurin + chemotherapy in newly diagnosed *FLT3*-WT AML	3	NCT03512197
	Crenolanib vs. midostaurin following induction chemotherapy and consolidation therapy in newly diagnosed *FLT3* mutant AML	3	NCT03258931
Quizartinib	Quizartinib with standard of care chemotherapy and as continuation therapy in new diagnosed *FLT3*-ITD AML	3	NCT02668653
	Quizartinib and venetoclax in r/r *FLT3* mutated AML	1b/2	NCT03735875
Crenolanib	Crenolanib combined with chemotherapy in r/r *FLT3* mutated AML	1b/2	NCT02298166
	Crenolanib maintenance following allogeneic stem cell transplantation in *FLT3*-mutant AML	2	NCT02400255
Gilteritinib	Gilteritinib vs. midostaurin in *FLT3* mutant AML during induction and consolidation chemotherapy	2	NCT03836209
	Gilteritinib as maintenance therapy following induction/consolidation therapy in *FLT3*-ITD AML in 1.CR	3	NCT02927262
Enasidenib/Ivosidenib	Ivosidenib or Enasidenib combined with induction/consolidation, followed by maintenance therapy in *IDH1/IDH2* mutated AML/MDS2-EB2	3	NCT03839771
	Enasidenib vs. conventional care regimens in *IDH2* mutant elderly AML	3	NCT02577406
	Ivosidenib vs. placebo in combination with azacitidine in *IDH1* mutant AML	3	NCT03173248
Glasdegib	Intensive chemotherapy +/− glasdegib or azacitidine +/− glasdegib AML patients	3	NCT03416179
	Immunotherapy combinations for AML for example, glasdegib plus avelumab	1b/2	NCT03390296
Venetoclax	Venetoclax combined with gilteritinib in r/r AML	I	NCT03625505
	Venetoclax +/− azacitidine in AML, ineligible for intensive treatment	3	NCT02993523
	Venetoclax +/− low dose cytarabine in AML, ineligible for intensive treatment	3	NCT03069352
	Venetoclax combined with induction/consolidation chemotherapy in AML	1b	NCT03709758

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
