# Peer review of "New Targeted Agents in Acute Myeloid Leukemia: New Hope on the Rise"

_ijms, 2019, doi:10.3390/ijms20081983_

Round 1

Reviewer 1 Report

The paper reports on novel agents for AML treatment. Even if it is enough up-to date, the major problem is the enciclopedic structure of it. Too many phase I study are reported, and the novel insights on new drugs are missing. No possible implication regarding the use of novel agents are reported, and the reader is confused from all these data.

Major comments:

1) the article should extensively reduced. I would strongly suggest to reduce the number of the data reported on each drug, and I would suggest to delete all phase I data, specially for drug who already got FDA approval. 

2) Speculation on new mechanisms of novel drugs are missing. There are no data on metabolic effects of Venetoclax (Pollyea et al, Nat Med 2018) in combo with HMAs. 

3) This paper is a list of clinical trials, without any speculation on how these novel agents will be integrated in clinical practice. 

4) References must be substantially revised. Several references do no have the name of the journal listed, some have abbreviations, other have the extended name of the Journal.

Author Response

Reviewer #1 (Comments to the Author):

We thank the reviewer for his thorough review our manuscript and his productive comments, which have addressed as detailed below:

The paper reports on novel agents for AML treatment. Even if it is enough up-to date, the major problem is the enciclopedic structure of it. Too many phase I study are reported, and the novel insights on new drugs are missing. No possible implication regarding the use of novel agents are reported, and the reader is confused from all these data.

Major comments:

1) the article should extensively reduced. I would strongly suggest to reduce the number of the data reported on each drug, and I would suggest to delete all phase I data, specially for drug who already got FDA approval. 

The focus of this manuscript is to provide a detailed overview of the current state of targeted treatment in AML. In our opinion it is essential for the readers to have a crude idea of the early clinical data since based on these findings further trials were set up and changed the clinical course for AML patients. We followed the reviewer´s suggestion and reduced the reported data in our manuscript (from initially 7543 words to 6333 words).

2) Speculation on new mechanisms of novel drugs are missing. There are no data on metabolic effects of Venetoclax (Pollyea et al, Nat Med 2018) in combo with HMAs. 

We agree with the reviewer on this point even though the required publication was already mentioned in our manuscript. Nevertheless we described the new mechanisms of the novel drugs combination more detailed.

Line 382-385

3) This paper is a list of clinical trials, without any speculation on how these novel agents will be integrated in clinical practice. 

As mentioned for comment number 1 the idea of this review is to up-date the readers about the current standard of targeted treatment and also to provide future perspectives, which we included in our manuscript. However, the focus of this review is rather current state of targeted therapies in AML than future perspectives and speculations being beyond the scope of this review.

4) References must be substantially revised. Several references do no have the name of the journal listed, some have abbreviations, other have the extended name of the Journal.

We agree with the reviewer and have made the required changes.

Reviewer 2 Report

This is a very nice review that summerizes the clinical trial results of various therapeutic strategies that are currently approved and tested. It is clearly written and also interesting since also upcoming agents are discussed. I have only a few minor comments.

Weird sentence at page 3 sentence 84 "This study did not just fail to show a clinical benefit for GO, but it was stopped early'' Explain why it is stopped early.

Page 5 sentence 169: two times demonstrated.

The various site effects like tumor lysis syndrome could be better eplained.

Author Response

Reviewer #2 (Comments to the Author):

We thank the reviewer for his thorough review our manuscript and his productive comments, which have addressed as detailed below:

This is a very nice review that summarizes the clinical trial results of various therapeutic strategies that are currently approved and tested. It is clearly written and also interesting since also upcoming agents are discussed. I have only a few minor comments.

Minor comments:

Weird sentence at page 3 sentence 84 "This study did not just fail to show a clinical benefit for GO, but it was stopped early'' Explain why it is stopped early.

We addressed the required changes and have explained in more detail why the study was stopped early.

Line 77-82

Page 5 sentence 169: two times demonstrated.

We have corrected this point.

The various site effects like tumor lysis syndrome could be better explained.

We have made the required changes.

Line 634-367

Round 2

Reviewer 1 Report

Nice job, Thank you for considering my objections and changing the manuscript accordingly.